# Dietary Supplementation with R-(+)-Limonene Improves Growth, Metabolism, Stress, and Antioxidant Responses of Silver Catfish Uninfected and Infected with *Aeromonas hydrophila*

**DOI:** 10.3390/ani13213307

**Published:** 2023-10-24

**Authors:** Elisia Gomes Da Silva, Isabela Andres Finamor, Caroline Azzolin Bressan, William Schoenau, Marina De Souza Vencato, Maria Amália Pavanato, Juliana Felipetto Cargnelutti, Sílvio Teixeira Da Costa, Alfredo Quites Antoniazzi, Bernardo Baldisserotto

**Affiliations:** 1Department of Physiology and Pharmacology, Universidade Federal de Santa Maria, Santa Maria 97105-900, RS, Brazil; elisia.silva@ufsm.br (E.G.D.S.); isabela.finamor@acad.ufsm.br (I.A.F.); caroline.azzolin@acad.ufsm.br (C.A.B.); william.schoenau@ufsm.br (W.S.); maria.amalia@ufsm.br (M.A.P.); alfredo.antoniazzi@ufsm.br (A.Q.A.); 2Department of Morphology, Universidade Federal de Santa Maria, Santa Maria 97105-900, RS, Brazil; marina.vencato@ufsm.br (M.D.S.V.); silvio.costa@ufsm.br (S.T.D.C.); 3Department of Preventive Veterinary Medicine, Universidade Federal de Santa Maria, Santa Maria 97105-900, RS, Brazil; juliana.cargnelutti@ufsm.br

**Keywords:** diet, limonene, *Aeromonas hydrophila*, *Igf1* expression, antioxidants, hepatoprotector

## Abstract

**Simple Summary:**

Increased intensification of the fish farming system requires care in production due to induced stress conditions, which contribute to the occurrence of infectious diseases. In this research, we evaluated how dietary supplementation with R-(+)-limonene, a compound found abundantly in essential oils from plants such as lemon and orange, affects growth, plasma and liver metabolic parameters, and antioxidant and stress responses in silver catfish, *Rhamdia quelen*, challenged or not with *Aeromonas hydrophila*. Dietary R-(+)-limonene (L0.5, L1.0, and L2.0 mL/kg of diet) improved the productive performance of fish and increased the expression of insulin growth factor 1 gene, which is related to growth. The metabolic responses indicated that the additive in the feed was not harmful to the health of the animals and increased resistance to *A. hydrophila*. R-(+)-limonene did not cause damage to the liver and exhibited a hepatoprotective effect against *A. hydrophila*. The expression of stress-related genes decreased with R-(+)-limonene dietary supplementation.

**Abstract:**

R-(+)-limonene is a monoterpene from plants of the genus *Citrus* with diverse biological properties. This research evaluated the effects of dietary supplementation with R-(+)-limonene on growth, metabolic parameters in plasma and liver, and the antioxidant and stress responses in silver catfish, *Rhamdia quelen*, challenged or not with *Aeromonas hydrophila*. Fish were fed for 67 days with different doses of R-(+)-limonene in the diet (control 0.0, L0.5, L1.0, and L2.0 mL/kg of diet). On the 60th day, a challenge with *A. hydrophila* was performed. R-(+)-limonene in the diet potentiated the productive performance of the fish. The metabolic and antioxidant responses indicate that R-(+)-limonene did not harm the health of the animals and made them more resistant to the bacterial challenge. Histological findings showed the hepatoprotective effect of dietary R-(+)-limonene against *A. hydrophila*. *Igf1* mRNA levels were upregulated in the liver of fish fed with an L2.0 diet but downregulated with bacterial challenge. The expression levels of *crh* mRNA were higher in the brains of fish fed with the L2.0 diet. However, the L2.0 diet downregulated *crh* and *hspa12a* mRNA expression in the brains of infected fish. In conclusion, the results indicated that R-(+)-limonene can be considered a good dietary supplement for silver catfish.

## 1. Introduction

Aquaculture has become the fastest growing food production technology in the world [1]. Allied to this expansion, care and control over the health and survival of animals also increase [2]. The practice of fish farming can lead to stressful situations [3], contributing to the occurrence of infectious diseases, with *Aeromonas hydrophila* being the most challenging pathogen associated with environmental stress [4,5].

The stress response results from several adjustments in physiological mechanisms in order to restore the animal’s homeostatic state and survival in suboptimal conditions such as low water quality and high density [6,7]. However, the difficulty in reestablishing the organism’s balance makes the animal prone to reduced productive performance, metabolic changes, inefficiency of the antioxidant defense system, and, in addition, increases mortality [7,8,9,10].

In the last decade, there has been a growing concern about the challenges faced by commercial fish farming. Different treatments and control options including immunostimulants such as glycoproteins, probiotics, macroalgae, and essential oils have been used to control diseases. Studies have shown that dietary supplementation of the glycoprotein lactoferrin increases the immune response of fish [11,12,13,14]. For example, lactoferrin supplements in the diet of Asian catfish (*Clarias batrachus*) increased the serum level of lysozyme and protection against *A. hydrophila* [15]. Diets containing macroalgae increased growth, lipid metabolism, and disease resistance in fish [16]. For example, seaweed (*Gracilaria pigmaea*) in the diet stimulates growth, antioxidant, and immunological status in Asian sea bass (*Lates calcarifer*) [17].

The use of essential oils (EO) and their phytochemicals in the dietary supplementation of fish is also promising as stimulators of animal growth, health, and well-being [18,19,20,21]. EOs can reduce oxidative stress, increase the immune response, and modulate the metabolic response of fish [7,22,23]. Several studies have shown that EOs from the genus *Citrus* improved the productive performance of common carp (*Cyprinus carpio*) [18], modulated the metabolic parameters of Nile tilapia (*Oreochromis niloticus*) [24] and sea bass (*Micropterus salmoides*) [25], increased the antioxidant capacity in silver catfish (*Rhamdia quelen*) [23] and tambaqui (*Colossoma macropomum*) [22], and increased the resistance and immune response in Mozambican tilapia (*Oreochromis mossambicus*) [26].

Blood biochemical parameters are markers commonly used in aquaculture and scientific research to evaluate the health and well-being of fish [27,28]. However, for a comprehensive assessment of the effects of various environmental or dietary factors, physiological analyses should be supplemented by histological ones [29,30,31,32].

Limonene is a monocyclic monoterpene found abundantly in EOs from citrus plants such as lemon and orange [33]. The isomer R-limonene or 4-isopropenyl-1-methylcyclohexene (C_10_H_16_) is considered the major compound in *Citrus* peels and the main active biological form of limonene [34,35]. Thus, the present work aimed to evaluate the effects of dietary supplementation with R-(+)-limonene on *R. quelen* challenged or not with *A. hydrophila*. These evaluations included a comprehensive analysis of fish growth parameters, blood plasma, and liver metabolic parameters, including lipid metabolism and the antioxidant system. Histology was performed on the liver, as it is a fundamental organ in metabolic processes. The expression of genes related to stress responses to infection challenges was also carried out in the brain.

## 2. Materials and Methods

### 2.1. Phytochemical

The phytochemical R-(+)-limonene (#W263303) was purchased from Sigma-AldrichTM (St. Louis, MO, USA).

### 2.2. Fish and Culture Conditions

Silver catfish (20 ± 0.6 g and 11.3 ± 0.4 cm) were purchased from a local fish farm and acclimatized for seven days in 71 L tanks, 15 fish per tank, with continuous aeration. During acclimatization, the fish were fed to satiation twice a day with commercial feed (Supra Juvenil, São Leopoldo, Brazil). Dissolved oxygen (7.88 ± 0.31 mg/L) and temperature (23.7 ± 0.5 °C) were measured daily with a Y 5512 oxygen meter (YSI, Yellow Springs, OH, USA). A water recycling system with mechanical and biological filtration was used. The levels of total ammonia (0.005 ± 0.001 mg/L) and nitrite (0.01 ± 0.01 mg/L) were checked daily using commercial kits (Labcon Test, Camboriú, Brazil), preserving water quality. As ammonia and nitrite levels were always low, nitrate levels were not measured. Feces and residues were removed every day. The water in the boxes was renewed by 20% every day. The Ethics on Animals Use Commission from the Federal University of Santa Maria approved all animal management procedures (#2005260821).

### 2.3. Diets

Diets were prepared according to Zeppenfeld et al. [36]. For this purpose, the ingredients were weighed, and during mixing, different doses of R-(+)-limonene were added (0-control, 0.5, 1.0, or 2.0 mL of R-(+)-limonene per kg of diet). After mixing the ingredients, the diets were moistened, pelleted, and taken to the oven to dry for 24 h at 50 °C. The composition of the standard diet is shown in Table 1. The silver catfish were fed twice a day, at 7:00 a.m. and 7:00 p.m., for 67 days, in an amount fixed at 5% of the biomass of each tank.

### 2.4. Challenge with A. hydrophila

The strain of *A. hydrophila* (MF 372510) was isolated from a naturally infected juvenile silver catfish and identified through biochemistry and molecular assays [37]. Silver catfish were inoculated intramuscularly with 0.1 mL of *A. hydrophila* solution (5.6 × 10^8^ CFU mL^−1^; OD_700_) on the right lateral dorsal side. Uninfected fish received the same dose of sterile saline through the same process. The challenge test was conducted for one week.

### 2.5. Experimental Design

For the experiment, the fish were divided into 8 groups in triplicate (*n* = 15 each replicate). The fish were fed with different doses of R-(+)-limonene for 67 days: control (0.0), 0.5 R-(+)-limonene (L0.5), 1.0 (L1.0), and 2.0 mL/kg of diet (L2.0). The diet was provided twice a day. Growth performance indices were evaluated at 30 and 60 days of treatment. On the 60th day, the animals were challenged with *A. hydrophila* or its vehicle for one week. After this period, the fish were anesthetized for 3 min with 300 μL/L of *Cymbopogon flexuosus* EO [38] for blood collection and then euthanized by sectioning the spinal cord to obtain liver and brain tissue.

### 2.6. Assays

#### 2.6.1. Growth Performance

The production parameters were evaluated through weight, standard length, weight gain (WG) = (final weight − initial weight), specific growth rate (SGR) = [(ln final weight − ln initial weight)/time] × 100, [39], feed conversion rate (FCR) = (feed intake/weight gain), [40], feed intake (FC) = (percentage of feed consumed per day), [41], hepatosomatic index (HSI) = (liver weight/body weight) × 100, and survival = (final number of fish/initial number of fish) × 100, [42]. The recorded mortality data were used to calculate the relative survival percentage (RSP), RSP = 1 − [(mortality (%) in the treated group)/(mortality (%) in the control group)] × 100 [43].

#### 2.6.2. Plasma Metabolic Parameters

The blood was centrifuged at 3500 rpm for 10 min at room temperature. Plasma was used to measure glucose, lactate dehydrogenase (LDH), and total protein, using kits (Gold Analisa, Belo Horizonte, Brazil) and lactate (BioTecnica, Varginha, Brazil).

#### 2.6.3. Liver Metabolic Parameters

Glucose and glycogen were determined according to Dubois et al. [44]. Lactate was evaluated as described by Harrower and Brown [45]. The protein content was measured according to Lowry [46].

#### 2.6.4. Lipid Profile

Plasma triglycerides and total cholesterol were measured using kits (Gold Analisa, Belo Horizonte, Brazil) and total lipid in liver tissue according to Frings et al. [47].

#### 2.6.5. Enzymatic Markers of Liver Damage

The activities of alanine aminotransferase (ALT) and aspartate aminotransferase (AST) were measured in the plasma, using kits (Gold Analisa^®^, Belo Horizonte, Brazil) and lactate (BioTecnica^®^).

#### 2.6.6. Antioxidant and Oxidative Parameters in the Liver

##### Superoxide Anion and Antioxidant Enzymes

Superoxide anion (O_2_^•−^) levels were measured as defined by Wang et al. [48], and the results were expressed as nmol min^−1^ g tissue^−1^. Total superoxide dismutase (SOD) was performed according to Mirsa and Fridovich [49], and the values were expressed as USOD g tissue^−1^. One SOD unit was defined as the amount of enzyme required for 50% inhibition of adrenochrome formation. Catalase (CAT) was evaluated as reported by Aebi [50], and the results were expressed as pmol g tissue^−1^.

##### Glutathione S-Transferase (GST)

Glutathione S-transferase (GST) activity, expressed as nmol min g tissue^−1^, was measured by following the rate of dinitrophenyl-S-glutathione formation at 340 nm as described by Habig et al. [51].

##### Oxidative Damage Biomarker

Lipid peroxidation was evaluated using the lipid hydroperoxide (LOOH) technique (LOOH) and performed as described by Hermes-Lima et al. [52]. LOOH results were expressed in nmol mg tissue^−1^.

##### Reduced and Oxidized Glutathione

Oxidized glutathione (GSSG) and total glutathione were estimated according to Giustarini et al. [53]. Reduced glutathione (GSH) levels were obtained by subtracting the measured amounts of GSSG (multiplied by 2, as in the recycling method one molecule of GSSG is reduced to two molecules of GSH) from the amounts of total glutathione. Results were expressed as nmol g tissue^−1^.

##### Total Antioxidant Capacity

The total antioxidant capacity (TAC) was verified according to Campos and Lissi [54], and the values were expressed in µmol g tissue^−1^.

#### 2.6.7. Histology

Fragments of liver tissue were carefully excised, washed with 0.9% NaCl, and fixed in 10% formalin buffer for 24 h. After fixation, the tissues were dehydrated in an alcohol series (70%, 80%, 90%, and 100%), then cleared with xylene solution, and finally embedded in paraffin at 56–58 °C. After embedding in paraffin, the molds were sectioned at 6 μm using an HM 325 rotary microtome (Thermo Scientific, Runcorn, UK). These tissue slides sets were stained with Goldner’s trichrome and then washed for microscopic analysis. The preparations were evaluated using an Axio Scope.A1 compound optical microscope (Zeiss, Jena, Germany) coupled to an Axiocam 105 digital color camera (Zeiss, Jena, Germany). For the morphometric analysis of the vacuoles, the images were divided into 12 quadrants of 20,000 µm^2^. A quantitative analysis was performed for vacuoles. To this end, the images were divided into forty-eight quadrants of 1000 μm^2^. ImageJ 1.54d, using the Grid plugin, was used to analyze these measurements.

#### 2.6.8. Gene Expression

The expression levels of genes directly related to stress, *hspa12a* (heat shock protein 70, member 12) and *crh* (corticotropin releasing hormone), were evaluated in the brain, and the *igf1* gene (insulin-like growth factor 1 (igf1)) in the liver (*n* = 8 per group for each of the expressed genes). Primer design and qPCR analysis were performed according to the protocol established by Souza et al. [55] and Da Silva et al. [56].

## 3. Statistical Analysis

The statistical analysis was made using the Statistica™ software version 11.0 (Statsoft, Tulsa, OK, USA). Prior to analysis, Bartletts’s and Levene’s tests were used to verify the normality and homogeneity of data distribution (*p* < 0.05), respectively. Almost all data presented homoscedasticity (Levene, *p* > 0.05) and a normal distribution, but values of glucose in the liver, triglycerides, and AST in the plasma were log transformed, and the square plasma was extracted from ALT values to attend these conditions. Growth performance was analyzed using one-way ANOVA, while data on metabolic, lipid, antioxidant, oxidative, molecular, and histological morphometry parameters were subjected to two-way ANOVA analysis (treatment × infected or not), each followed by Tukey’s multiple comparison test (*p* < 0.05). The data from *igf1* expression were not parametric (Levene, *p* < 0.05), so they were analyzed by the Kruskal–Wallis/Scheirer–Ray–Hare test, followed by the Nemenyi test. Results are reported as mean ± SEM, and differences were considered significant when *p* < 0.05. For the analysis of growth, metabolism, and lipid profile, three fish from each replicate were used (*n* = 9). For the remainder of the analyses, three fish from two replicates (randomly chosen) and two fish from the other were used (*n* = 8).

## 4. Results

External signs indicated 100% of the fish from the control and 25% of those fed diets L0.5 and L1.0 were infected with the bacterium. These fish presented ulcers with reddish edges at the inoculation site, hemorrhage in the fins, lesions on the tail, and shortening of the barbels. Fish fed diet L2.0 did not present any lesions or hemorrhage, only reddish fins.

### 4.1. Growth Performance

Fish fed with L0.5 and L2.0 diets showed higher weight and SGR at 30 and 60 days of treatment compared to the fish fed with the control diet (*p* < 0.05). The L0.5 and L2.0 diets exhibited a lower FCR than fish fed with control and L1.0 diets at 60 days of treatment (*p* < 0.05). There was no mortality during all the experiments (Table 2).

### 4.2. Plasma Metabolic Parameters

There was a significant interaction between diet and infection for glucose, lactate, total proteins, and LDH (*p* < 0.05), but diets did not significantly affect these parameters in uninfected fish (*p* > 0.05). All the groups treated with R-(+)-limonene and challenged with *A. hydrophila* had lower glucose, lactate, glycogen, and total protein in plasma in relation to their infected control group. The L0.5 group infected with *A. hydrophila* presented higher lactate levels than the L0.5 non-infected group (*p* < 0.05) and the L1.0 and L2.0 infected groups (*p* < 0.05). The L2.0 group infected with *A. hydrophila* showed higher lactate levels than the non-infected L2.0 group (*p* < 0.05). Moreover, LDH levels were lower in the L1.0 group infected with *A. hydrophila* than in the infected control and L1.0 non-infected groups (*p* < 0.05) (Table 3).

### 4.3. Liver Metabolic Parameters

There was a significant interaction between diet and infection for lactate, glycogen, and total protein levels, but the diets did not significantly change these hepatic metabolites in uninfected fish (*p* > 0.05) in relation to the control group, except for the higher glucose level in the L2.0 group compared to the uninfected control and L0.5 groups (*p* < 0.05) (Table 3).

Glucose, lactate, and glycogen levels were all lower in the infected groups treated with R-(+)-limonene in relation to their infected control group. The L1.0 group infected with *A. hydrophila* exhibited lower lactate levels than L0.5 infected with *A. hydrophila* (*p* < 0.05), which presented higher lactate levels than the L0.5 non-infected group (*p* < 0.05) (Table 3). Otherwise, total protein in the infected groups treated with R-(+)-limonene were all higher than their infected control group (*p* < 0.05) (Table 3).

Glycogen levels in fish challenged with *A. hydrophila* were higher than in the non-infected control group (*p* < 0.05). However, when fed with the L0.5, L1.0, and L2.0 diets, the glycogen levels were significantly lower than in the control infected with *A. hydrophila*, but still higher than their respective non-infected groups (*p* < 0.05) (Table 3).

### 4.4. Lipid Profile

There was a significant interaction between diet and infection for total lipid levels in the liver (*p* < 0.05) but not for plasma triglycerides and cholesterol (*p* > 0.05). Fish fed the L1.0 and L2.0 diets significantly reduced plasma triglycerides levels compared to the control group and L0.5 in uninfected fish (*p* < 0.05). The diets did not significantly affect plasma cholesterol levels in uninfected fish (*p* > 0.05). Non-infected fish fed the L0.5 diet presented the lowest total lipid levels in the liver compared to the control, L1.0, and L2.0 uninfected groups (*p* < 0.05) (Table 4).

All fish groups infected with *A. hydrophila* and treated with R-(+)-limonene exhibited significantly lower plasmatic levels of triglycerides and cholesterol in relation to both controls (non-infected infected with *A. hydrophila*) (*p* < 0.05). In addition, the L0.5 group infected with *A. hydrophila* showed lower triglyceride levels than the non-infected L0.5 group (*p* < 0.05) (Table 4).

The L1.0 and L2.0 groups infected with *A. hydrophila* exhibited lower levels of total lipid than the control and L0.5 groups infected with *A. hydrophila*, who presented higher total lipid levels than the non-infected control group (*p* < 0.05). Moreover, fish challenged with *A. hydrophila* and fed the L0.5 diet showed higher levels of total lipid than the non-infected L0.5 group (*p* < 0.05) (Table 4).

### 4.5. Liver Transaminases Activity

There was a significant interaction between diet and infection for ALT and AST activities (*p* < 0.05). Fish fed the L0.5 and L2.0 diets significantly reduced ALT activity compared to the control and L1.0 groups in uninfected fish (*p* < 0.05); however, diets did not significantly affect AST activity in uninfected fish (*p* > 0.05) (Figure 1A,B).

There were no significant differences in AST activity in all the non-infected groups. All the infected groups treated with limonene showed a reduction in AST activity in relation to their control group (*p* < 0.05). AST activity in the L0.5 group infected with *A. hydrophila* was significantly lower than in the non-infected L0.5 and control groups (*p* < 0.05) (Figure 1A,B).

### 4.6. Superoxide Anion and Antioxidant Enzymes

Diets did not significantly affect O_2_^•−^ levels in uninfected fish (*p* > 0.05), but there was a significant interaction between diet and infection. O_2_^−^ levels were significantly higher in control fish challenged with *A. hydrophila* compared to the uninfected control group (*p* < 0.05). In infected fish fed the L0.5, L1.0, and L2.0 diets, O_2_^•−^ levels were significantly lower than their infected control group (*p* < 0.05) (Figure 2A).

There was a significant interaction between diet and infection for SOD activity. Fish fed the diets L1.0 and L2.0 significantly increased SOD activity compared to the control group and L0.5 in uninfected fish (*p* < 0.05). The control fish infected with *A. hydrophila* showed higher SOD activity than the non-infected control (*p* < 0.05). The L0.5, L1.0, and L2.0 infected groups exhibited significantly higher SOD activity than the non-infected and infected controls (*p* < 0.05). Infected fish fed the L1.0 and L2.0 diets presented higher SOD activity than the control and L0.5 groups infected with *A. hydrophila* (*p* < 0.05). L0.5 and L2.0 infected groups showed higher SOD activity than the L0.5 and L2.0 non-infected groups (*p* < 0.05) (Figure 2B).

There was a significant interaction between diet and infection for CAT activity. The diets significantly increased CAT activity compared to the control group in uninfected fish (*p* < 0.05). The control, L1.0, and L2.0 groups infected with A. hydrophila exhibited lower CAT activity compared to the L0.5 diet (*p* < 0.05) in infected fish. Fish infected with A. hydrophila and those infected and fed L1.0 and L2.0 diets showed lower CAT activity than the uninfected control group (*p* < 0.05) (Figure 2C).

### 4.7. Glutathiones

Diets did not significantly affect GST activity in uninfected fish (*p* > 0.05), but there was a significant interaction between diet and infection. The infected control showed lower enzymatic activity compared to the uninfected control (*p* < 0.05). The L0.5 group infected with *A. hydrophila* showed higher GST activity compared to the other infected groups (*p* < 0.05) (Figure 3A).

There was a significant interaction between diet and infection in GSH levels. Fish fed the L2.0 diet had significantly increased GSH levels compared to control, L0.5, and L1.0 in uninfected fish (*p* < 0.05). The GSH levels were lower in infected fish and in those infected and fed with L0.5 and L1.0 diets compared to the non-infected control (*p* < 0.05). The infected L2.0 group showed significantly higher GSH levels compared to the control infected with *A. hydrophila* (*p* < 0.05), whereas fish infected and fed the L0.5, L1.0, and L2.0 diets exhibited lower GSH levels compared to their respective uninfected groups (*p* < 0.05) (Figure 3B).

Diets did not significantly affect GSSG levels in uninfected fish (*p* > 0.05), but there was a significant interaction between diet and infection. GSSG levels were higher in infected control fish than in uninfected control fish (*p* < 0.05). Infected fish fed the L0.5, L1.0, and L2.0 diets showed significantly reduced GSSG levels compared to the infected control group (*p* < 0.05) (Figure 3C).

### 4.8. Oxidative Damage Biomarker

Diets did not significantly affect LOOH levels in uninfected fish (*p* > 0.05), but there was a significant interaction between diet and infection. LOOH levels were higher in fish challenged with *A. hydrophila* than in the uninfected control (*p* < 0.05), although they were lower in infected fish fed the L0.5 and L2.0 diets compared to the control group (*p* < 0.05) (Figure 4). 

### 4.9. Total Antioxidant Capacity

There was a significant interaction between diet and infection on TAC levels. Fish fed the L0.5 diet had significantly increased TAC levels compared to the control group, L1.0, and L2.0 in uninfected fish (*p* < 0.05). TAC levels were lower in infected control fish than in uninfected control fish (*p* < 0.05). Infected fish fed the L0.5 diet exhibited significantly higher TAC levels than the other infected groups (*p* < 0.05) (Figure 5).

### 4.10. Histology

Vacuolation and lipid deposits in hepatocytes, nuclear pyknosis, binucleated hepatocytes, erythrocyte infiltration in the sinusoids, fibrosis, and hemorrhage were evaluated histopathologically (Figure 6). Uninfected fish fed the control, L0.5, L1.0, and L2.0 diets did not show any histological changes in the liver (Figure 6A–D). Control fish challenged with *A. hydrophila* showed intense vacuolization of hepatocytes, infiltration of erythrocytes in the blood sinusoids, nuclear pyknosis, fibrosis, and hemorrhage (Figure 6E). Erythrocyte infiltration in the blood sinusoids and nuclear pyknosis were observed in fish fed the L0.5 diet (Figure 6F). Fish fed the L1.0 diet exhibited vacuolization of hepatocytes and binucleated hepatocytes (Figure 6G). Infiltration of erythrocytes in the blood sinusoids was observed in fish fed the L2.0 diet (Figure 6H).

Diets did not significantly affect the number of vacuoles in uninfected fish (*p* > 0.05), but there was a significant interaction between diet and infection. The number of vacuoles in fish infected with *A. hydrophila* was greater than in the uninfected control group (*p* < 0.05). On the other hand, the infected groups fed the L0.5 and L2.0 diets showed a reduction in the number of vacuoles compared to the infected control group (*p* < 0.05) (Figure 7A).

There was a significant interaction between diet and infection on the size of vacuoles. Fish fed the L1.0 diet significantly increased the size of vacuoles in the hepatocytes compared to control, L0.5, and L2.0 uninfected groups (*p* < 0.05). The number of vacuoles in fish infected with *A. hydrophila* was greater than in the uninfected control group (*p* < 0.05). On the other hand, the infected groups fed the L0.5 and L2.0 diets showed a reduction in the number of vacuoles compared to the infected control group (*p* < 0.05) (Figure 7B).

### 4.11. Gene Expression

There was a significant interaction between diet and infection on *igf1* mRNA levels in the liver. The L2.0 diet significantly increased *igf1* mRNA levels in the liver compared to the control, L0.5, and L1.0 groups in uninfected fish (*p* < 0.05). Challenge with *A. hydrophila* reduced *igf1* mRNA levels in all groups compared to the uninfected control (*p* < 0.05). *Igf1* mRNA levels were lower in the infected L2.0 group than in the uninfected L2.0 group (*p* < 0.05) (Figure 8A).

Diets did not significantly affect brain *hspa12a* mRNA levels in uninfected fish (*p* > 0.05), but there was a significant interaction between diet and infection. Infected fish fed the L2.0 diet showed downregulation of *hspa12a* mRNA expression in relation to the infected control and those fed the L0.5 diet (*p* < 0.05). The L0.5 diet in infected fish stimulated higher levels of *hspa12a* mRNA than the control and L0.5 diets in uninfected fish (*p* < 0.05) (Figure 8B).

There was a significant interaction between diet and infection on *crh* expression. In infected fish, the L2.0 diet significantly decreased *crh* expression in relation to the infected control and the L0.5 diet (*p* < 0.05). The expression of *crh* mRNA levels in infected fish fed the L2.0 diet was lower than in uninfected fish fed the L2.0 diet (*p* < 0.05) (Figure 8C).

## 5. Discussion

Phytochemicals are compounds with numerous bioactivities that have been continuously used in the pharmaceutical industry and the scientific community [57]. EO from the genus *Citrus*, with R-(+)-limonene as the main compound, has several therapeutic effects in fish, including growth-promoting and antioxidant and anti-inflammatory properties [22,23,58,59].

This study showed that supplementation of R-(+)-limonene in the diet improved the productive performance of fish, increasing weight and specific growth rate, in addition to improving the feed conversion rate. Metabolic responses indicated that R-(+)-limonene was not harmful to the health of the animals and increased resistance to the challenge with *A. hydrophila*, the main pathogen that causes disease in fish. Antioxidant defenses were stimulated, and the lowest concentration of R-(+)-limonene showed better performance against the pathogen. Histological findings of liver tissue showed that nutrition with R-(+)-limonene did not cause liver damage and that it has a hepatoprotective effect against *A. hydrophila*. Expression of the growth-related gene increased with the highest concentration of R-(+)-limonene in the diet but decreased when challenged with *A. hydrophila*. R-(+)-limonene in the diet of fish with *A. hydrophila* inhibited the expression of genes related to stress.

Dietary supplementation with R-(+)-limonene enhanced catfish production parameters, with an increase in final weight and TCE at 30 and 60 days and better feed conversion. According to Gültepe [60], good growth performance can be attributed to the rapid absorption of EOs in the gastrointestinal tract of fish. The current results are consistent with data found in Mozambican tilapia fed *C. sinensis* EO (83% limonene), whose final weight and SGR were higher than those fed a controlled diet [26]. Ngugi et al. [61] observed that the addition of *Citrus limon* peel EO (81% limonene) to the diet of *Labeo victorianus*, in addition to weight and TCS, changed feed conversion.

Changes in metabolic parameters in fish blood have been used as a reference to understand the effects of diets on animal metabolism and health [62,63]. Glucose, lipids, and proteins are the main sources of energy for fish. The determination of total proteins in plasma is a sensitive and physiologically relevant parameter, and when elevated in plasma, it can be considered together with other factors useful for signaling the health and immunity status of fish [64]. The mobilization of plasma proteins increases according to the intensity of stress to meet energy demand, maintain osmoregulation, and increase the immune response [65,66]. Glucose, an important nutrient for energy, varies according to the physiological state of the animal [62]. Increased plasma glycemia may also be associated with stressful situations or illnesses. Lactate, an important biomarker of stress, is an end product of the glycolysis process, formed by the conversion of pyruvate through the enzyme LDH (terminal enzyme of anaerobic glycolysis) [67]. Elevated lactate levels suggest the use of anaerobic pathways to obtain energy in the absence of oxygen from the environment, or functional pathways, when fish cannot capture it [68]. Changes in LDH activity can arise when tissues are infected or damaged, in addition to indicating an imbalance in anaerobic carbohydrate metabolism [69]. Considering that plasma metabolic parameters were not altered, it can be assumed that dietary supplementation with different doses of R-(+)-limonene did not stress or harm the catfish health. Furthermore, faced with the challenge of *A. hydrophila*, R-(+)-limonene in the diets made the fish more resistant to damage caused by the bacteria. Ngugi et al. [61] reported that concentrations of 5 and 8% (50 and 80 mL per kg of diet, respectively) *C. limon* EO in the diet of *L. victorianus* infected with *A. hydrophila* increased fish resistance, since levels of glucose and total proteins remained similar to the levels of healthy fish. Partially corroborating our data, Gültepe et al. [70] found that R-(+)-limonene derived from orange peel EO and supplemented in the diet of rainbow trout increased resistance against *Yersinia ruckeri*, since serum glucose and total protein levels did not change. However, contrary to our data, the addition of 1 mL of R-(+)-limonene per kg of diet induced an increase in LDH in rainbow trout infected with *Y. ruckeri*.

The liver was chosen as the focus of this research because it is a key organ in metabolic processes as it is important for triggering the body’s defenses with metabolic readjustments after being challenged with pathogens [71]. This organ is essential in the synthesis of plasma proteins and endocrine factors, such as insulin-like growth factors (IGFs), and therefore influences, in addition to metabolism, the growth of the entire organism. It is also considered the main site of detoxification [72] and most susceptible to toxicant-induced injury [73,74,75,76].

Most physiological processes, such as growth, are modulated by the endocrine system. *Igf1*, produced in the liver, is one of the main hormones involved in regulating fish growth [77,78,79]. *Ifg1* expression levels are often compared with growth in length or weight [79]. Studies have reported that *igf1* mRNA levels in the fish liver can be used as a growth index marker [80,81,82]. Consistent with the production performance results, fish fed the highest dietary dose of R-(+)-limonene exhibited higher levels of hepatic *igf1* mRNA expression, suggesting that this phytochemical stimulated fish growth. A study conducted by Aanyu et al. [78] showed an increase in the expression of *igf1* mRNA in the liver tissue of Nile tilapia fed with the highest concentration of limonene (600 ppm) tested in diets. According to these same authors, one of the possible mechanisms by which dietary limonene stimulates growth is through increasing the energy availability of feed ingredients. Conversely, *igf1* mRNA levels were reduced following *A. hydrophila* infection in fish fed all the diets. Similar results were observed when different fish species were infected with distinct types of pathogens [83,84], including *A. hydrophila* [85].

The liver synthesizes and stores glucose in the body in the form of glycogen, an important energy reserve polysaccharide for animals [86]. Thus, the synthesis and degradation of hepatic glycogen are regulated to maintain plasma glucose levels, which are essential for the body’s needs [87]. The results of metabolic parameters in the liver showed an increase in glucose levels with the highest dietary dose of R-(+)-limonene; however, glycogen, lactate, and protein levels were not changed. This demonstrates that the animals were physiologically healthy since it was not necessary to activate the hepatic energy homeostasis mechanisms. In contrast to our results, the lowest concentration (0.25 mL EO kg/diet) of *Citrus* × *latifolia* EO in the tambaqui diet increased hepatic glucose levels [22]. Likewise, Lopes et al. [23] found that adding *Citrus aurantium* EO to the silver catfish diet did not alter liver glycogen levels; however, it did cause an increase in lactate and protein levels.

Stress caused by bacteria induces the animal’s body to initiate metabolic reprogramming in order to modulate the energy deficiency resulting from stress, mainly in the liver, which is a fundamental organ in adapting to stress. It can be seen that in the challenge with *A. hydrophila*, dietary R-(+)-limonene prevented a hyperglycemic condition caused by the bacteria in silver catfish. Carbohydrate metabolism is significantly affected by primary and secondary stress responses [88,89,90]. Lin et al. [91] stated that blood hyperglycemia to meet energy demands in stressful situations is linked to hepatic glycogenolysis. In this study, dietary R-(+)-limonene prevented glycogen degradation in catfish infected with *A. hydrophila*, as plasma glucose parameters remained normal. Furthermore, it preserved hepatic protein metabolism and prevented protein catabolism in silver catfish infected with *A. hydrophila*, since plasma protein levels were normal.

Lipids are important structural elements of the cell membrane, with functions in cell signaling and inflammation. Changes in the lipid profile may be related to the symptoms of the disease, particularly in the case of bacterial infections [92]. According to Dillard and German [93], terpenes are effective in reducing plasma cholesterol levels. Vieira et al. [94] and Ahmad and Beg [95] reported that diets containing limonene reduced, in addition to total cholesterol, serum triglyceride levels. The results of our investigation are consistent with these studies in that dietary R-(+)-limonene reduced triglyceride levels and did not alter basal cholesterol levels. Against *A. hydrophila*, all diets supplemented with R-(+)-limonene were effective in reducing triglyceride and cholesterol levels. In relation to total lipids, diets containing R-(+)-limonene induced a reduction in lipid content in the liver of fish, and when fish were challenged with *A. hydrophila*, diets with R-(+)-limonene managed to return the values of this parameter to control levels. Similar to our results, Baba et al. [20], using EO from *Citrus limon*, and Acar et al. [26], using EO from *C. sinensis*, observed that diets supplemented with *citrus* oils decreased the levels of these parameters in tilapia. It is possible to note in this research that R-(+)-limonene promoted hypocholesterolemic and hypolipidemic effects in silver catfish challenged with *A. hydrophila*.

Liver function transaminases (ALT and AST) are important enzymes in the mobilization of amino acids for gluconeogenesis and are associated with liver disease. They are used to monitor the function and integrity of liver tissue and damage to fish health, and high levels are associated with liver damage [96]. The results of this investigation point to a hepatoprotective effect of R-(+)-limonene, an indication that this phytochemical helps maintain normal function by accelerating the regenerative capacity of liver cells. In uninfected fish, R-(+)-limonene decreased ALT activity and preserved the activity of control AST, suggesting that liver tissue did not suffer any damage from R-(+)-limonene supplementation. Diets supplemented with R-(+)-limonene protected the liver from damage caused by *A. hydrophila*, as it inhibited the increase in ALT and AST activity. Sadeghi et al. [97], when using *C. aurantifolia* EO as a dietary supplement for common carp infected with *A. hydrophila*, noted that the tested additive did not harm liver function, since ALT and AST activities did not increase. These results suggest that dietary doses of R-(+)-limonene did not have a negative impact on liver function. Together with the stable values of triglycerides, total plasma cholesterol, and total lipids in the liver, it can be inferred that R-(+)-limonene is recommended to maintain the health of silver catfish. Studies have also reported that changes in protein and carbohydrate metabolism due to stressful situations lead to changes in ALT and AST activity [74]. However, in this investigation, the levels of glucose and total plasma and liver proteins in healthy fish and with *A. hydrophila*, treated with R-(+)-limonene, presented control values.

The balance between oxidation and antioxidant systems can be affected by stress, which causes excessive formation of reactive oxygen species (ROS) or reduced antioxidant production [98]. Endogenous antioxidant activity is essential to protect the organism from oxidative damage caused by ROS [99] and antioxidant enzymes, such as SOD, CAT, and glutathione peroxidase, which are the first line of protection. In fish, these enzymes are capable of deactivating the harmful effects of ROS and of modulating and restoring antioxidant mechanisms [100]. While SOD converts the O_2_^•−^ radical into hydrogen peroxide, CAT is responsible for eliminating it. A failure in this elimination will result in LPO that can be converted into an excretable molecule by GST [101,102]. In this study, fish fed diets supplemented with R-(+)-limonene exhibited greater SOD and CAT antioxidant activities, suggesting that this phytochemical can increase the capacity of the animal’s enzymatic antioxidant system. Similar results in increasing the activities of these enzymes were observed by Mohamed et al. [58] and Rahman et al. [59] in Nile tilapia fed diets containing EO from *C. sinensis* and African catfish (*Clarias gariepinus*) fed diets containing EO *from C. limon*, respectively. In the present study, O_2_^•−^ and LOOH levels and GST activity were not changed with the addition of R-(+)-limonene to the diet. This suggests that limonene in the diet is non-toxic, does not cause stress and lipoperoxidation to the liver, and does not impair antioxidant defenses. The results of GST activity differ from Lopes et al. [23], whose diets supplemented with *Citrus* × *aurantium* EO decreased the activity of this enzyme. GSH, the main endogenous antioxidant, is found in the thiol-reduced form (GSH) and the disulfide-oxidized form (GSSG) [103], acting to detoxify ROS and maintain redox balance [104]. Oxidative stress in fish can be detected by the depletion of GSH and the increase of LOOH in liver tissue [105]. As noted, stress can be avoided, as the liver did not show lipid peroxidation and GSH content increased in fish fed the highest dose of R-(+)-limonene, similar to Nile tilapia, whose diets contained EO *C. limon* [59]. However, GSSG levels in healthy fish were not altered by the R-(+)-limonene diet. Total antioxidant capacity can be considered the ability of a compound to act by reducing or neutralizing pro-oxidants [106,107], and its measurement is important to evaluate the protective activities of all antioxidants present in a sample [108]. In this study, the lowest dose of R-(+)-limonene in the diet stimulated antioxidant responses by increasing TAC. Likewise, Salem et al. [98] found that sea bream (*Sparus aurata*) fed orange peel in the diet showed a better antioxidant response with increased TAC activity.

*Aeromonas hydrophila* infection can induce changes in antioxidant defenses [109], suggesting that this change may be caused by increased O_2_^•−^ levels in combination with decreased SOD activity. A study with brown trout (*Salmo trutta trutta*) showed that infection with *A. hydrophila* reduced the activity of SOD and CAT enzymes [110]. In this investigation, we observed significant interactions between the effects of dietary supplementation with R-(+)-limonene and *A. hydrophila* infection on O_2_^•-^ levels and hepatic SOD activity. Diets with three doses of R-(+)-limonene decreased O_2_^•-^ levels and increased SOD activity in the liver of fish challenged with *A. hydrophila*. On the other hand, the lowest dose of R-(+)-limonene in the diet of fish infected with *A. hydrophila* increased CAT activity. These results show that R-(+)-limonene in the diet protected the catfish liver in the face of challenging situations, such as *A. hydrophila* infection, and increased the activity of the first defense enzymes (SOD and CAT). Apparently, the aggressive effect of *A. hydrophila* on an insufficient antioxidant system causes the activation of lipid peroxidation; therefore, tissue LOOH levels are elevated [111]. However, in this study, LOOH levels were markedly reduced in infected fish fed diets containing R-(+)-limonene, while GST activity returned to control values only with the L0.5 diet. Although GST activity in the other doses of R-(+)-limonene in the diet was reduced in relation to the control diet, it is likely that this enzyme was not necessary for detoxification, since the lipoperoxidation products were reduced. Similar findings were observed in silver catfish treated with caffeine-supplemented diets, whose treatment prevented the increase in hepatic LOOH levels and prevented the inhibition of SOD and GST activities caused by *A. hydrophila* [112]. The addition of R-(+)-limonene to the diets of fish with *A. hydrophila* did not prevent the reduction of GSSG levels in relation to infected and untreated fish, while the lowest dose of R-(+)-limonene increased the CT scan. Therefore, these results indicate that in the face of an *A. hydrophila* infection, the addition of R-(+)-limonene stimulates antioxidant activity, reduces tissue damage, and increases resistance to oxidative stress, improving the health status of silver catfish. Swamy et al. [113] observed that enzymatic antioxidant activity and resistance against *A. hydrophila* improved when Nile tilapia were fed diets supplemented with bay leaf *(Laurus nobilis*), which has limonene as one of its constituents.

Lesions in tissue and cellular morphology are useful in clarifying pathological processes and in identifying and diagnosing different diseases [18,114]. The fish liver is an organ sensitive to damage caused by *A. hydrophila*. As expected, the histopathological results of *A. hydrophila* infection in untreated silver catfish showed distinct morphological and quantitative changes in the liver tissue. An increase in the number and size of vacuoles in hepatocytes, erythrocyte infiltration in the sinusoids, hemorrhage, fibrosis, and nuclear pyknosis were observed. Silver catfish fed the lowest and highest dose of the phytochemical in the diet exhibited erythrocyte infiltration in the blood sinusoids; in addition, the former also presented nuclear pyknosis. In fish fed the L1.0 diet, vacuolation of hepatocytes and binucleated hepatocytes were observed. The lowest dose of R-(+)-limonene in the diet did not prevent the increase in the number of vacuolation in hepatocytes; however, it prevented their growth. Mustahal et al. [115] observed that catfish (*Clarias* sp.) infected with *A. hydrophila* and not treated with a probiotic-based diet presented liver congestion and lipid accumulation in the liver due to the formation of vacuoles, which according to Kalaiyarasi et. al. [116], compromises the lipid metabolism of fish. In another study, carried out by Al Mamun et al. [117], *A. hydrophila* infection in catfish (*Pangasianodon hypophthalmus*) caused necrosis of hepatocytes and pyknotic nuclei. The cause of hemorrhage and fibrosis in the liver tissue of Nile tilapia is associated with the release of toxins and extracellular products produced by *A. hydrophila* [118]. Our results showed that R-(+)-limonene supplementation in the diet of fish with *A. hydrophila* had a protective effect on liver tissue, as it reduced the number and size of vacuoles in hepatocytes. A hepatoprotective effect was observed in Nile tilapia fed with clove, basil, and ginger EO after being challenged with *Streptococcus agalactiae* [119].

Heat shock proteins (HSP) are molecules that can be activated in stress situations and in the immune response in bacterial infections, acting to protect proteins against denaturation caused by stressors [120]. In aquaculture, HSPs are important in signaling the development of inflammation and the immune response to pathogens [121,122]. In this investigation, in the challenge with *A. hydrophila*, the highest dose of R-(+)-limonene in the fish diet induced a decrease in the expression of *hspa12a*, and the lowest dose did not prevent the increase in the expression of this gene. The higher the expression of *hspa12a*, the greater the stressor capacity of *A. hydrophila* infection in silver catfish [121]. In fish, the stress response mediated by the hypothalamic–pituitary–interrenal (HPI) axis is associated with CRH [123,124]. The HPI axis, when activated by a stressor, releases CRH from neurons in the preoptic neurosecretory area [123], stimulating the pituitary gland to secrete adrenocorticotropic hormone (ACTH) [125]. ACTH, in turn, stimulates interrenal cells to produce and secrete corticosteroids, such as cortisol. This corticosteroid acts on energy metabolism, resulting in an increase in serum glucose levels, in order to meet the energy needs of the stressed animal [124]. In the present study, the highest dose of R-(+)-limonene induced an increase in the expression of *crh* in the brain, but in the challenge with *A. hydrophila*, the diet with the phytochemical decreased the expression of this gene. However, low dietary doses of R-(+)-limonene did not prevent the stress caused by the bacteria, indicating the activation of the HPI pathway, as it increased the expression of *crh*. Similar data were found by Bandeira Junior et al. [121], when they found that the effect of *A. hydrophila* infection increased the expression of stress-related genes (*hspa12a* and *crh*) in the brain.

## 6. Conclusions

In conclusion, the current research recommends the consumption of diets containing 0.5% R-(+)-limonene in their composition to increase the productive performance of silver catfish and improve their well-being even in the face of a challenging situation such as infection with *A. hydrophila*, not only modulating metabolic responses and reducing stress, but also protecting the liver against morphological changes and oxidative stress.

## Figures and Tables

**Figure 1 animals-13-03307-f001:**
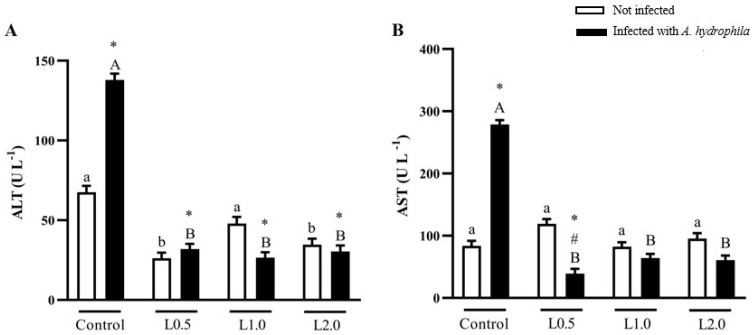
Liver transaminases, alanine (ALT) (**A**) and aspartate aminotransferase (AST) (**B**), activities in plasma of *Rhamdia quelen* fed different doses of R-(+)-limonene in the diet and infected or not with *A. hydrophila*. Data are presented as mean ± SEM (*n* = 8). Two-way ANOVA and Tukey’s test, *p* < 0.05. Different lowercase letters indicate significant differences between healthy groups. Different capital letters indicate a significant difference between groups infected with *A. hydrophila*. The number sign (#) indicates a significant difference between the groups infected with *A. hydrophila* in relation to the healthy groups. The asterisk (*) indicates a significant difference between the groups infected with *A. hydrophila* in relation to the respective healthy control group.

**Figure 2 animals-13-03307-f002:**
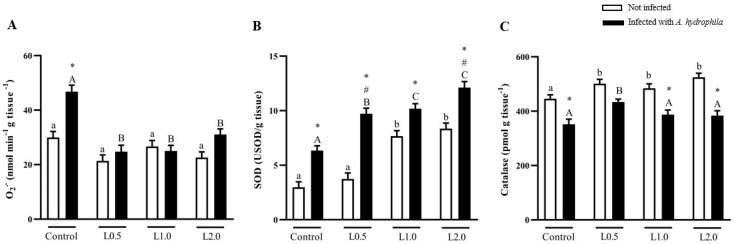
Superoxide (O_2_^•−^) levels (**A**) and superoxide dismutase (SOD) (**B**) and catalase (CAT) (**C**) activities in the liver of *Rhamdia quelen* fed different doses of R-(+)-limonene in the diet and infected or not with *A. hydrophila*. Data are presented as mean ± SEM (*n* = 8). Two-way ANOVA and Tukey’s test, *p* < 0.05. Different lowercase letters indicate significant differences between healthy groups. Different capital letters indicate a significant difference between groups infected with *A. hydrophila*. The number sign (#) indicates a significant difference between the groups infected with *A. hydrophila* in relation to the healthy groups. The asterisk (*) indicates a significant difference between the groups infected with *A. hydrophila* in relation to the respective healthy control group.

**Figure 3 animals-13-03307-f003:**
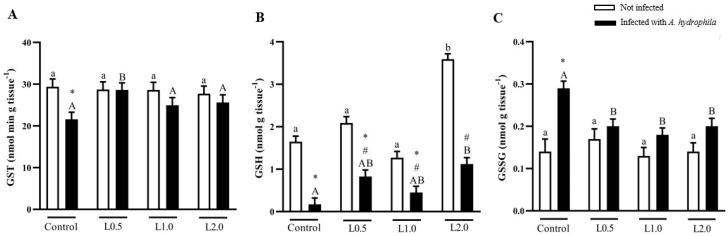
Glutathione S-transferase (GST) activity (**A**), reduced (GSH) (**B**), and oxidized glutathione (GSSG) (**C**) levels in the liver of *Rhamdia quelen* fed different doses of R-(+)-limonene in the diet and infected or not with *A. hydrophila*. Data are presented as mean ± SEM (*n* = 8). Two-way ANOVA and Tukey’s test, *p* < 0.05. Different lowercase letters indicate significant differences between healthy groups. Different capital letters indicate a significant difference between groups infected with *A. hydrophila*. The number sign (#) indicates a significant difference between the groups infected with *A. hydrophila* in relation to the healthy groups. The asterisk (*) indicates a significant difference between the groups infected with *A. hydrophila* in relation to the respective healthy control group.

**Figure 4 animals-13-03307-f004:**
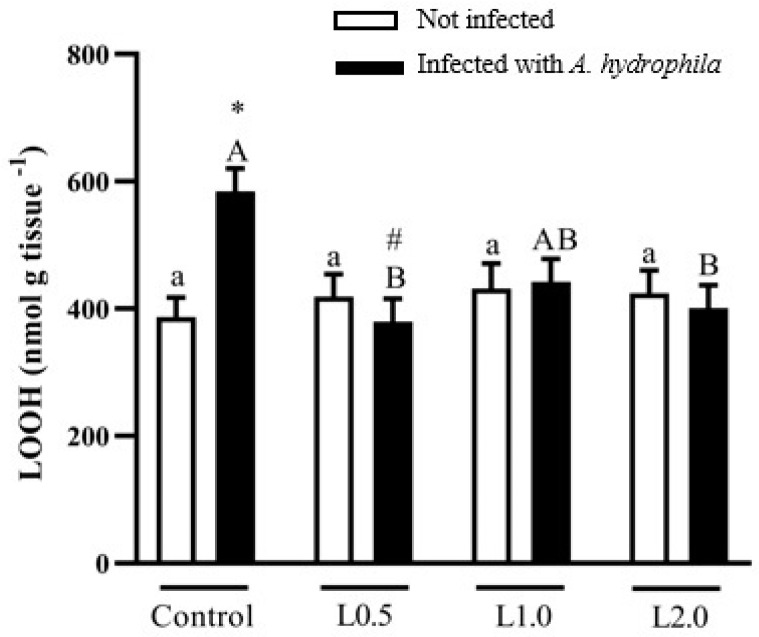
Lipid hydroperoxide levels in the liver of *Rhamdia quelen* fed different doses of R-(+)-limonene in the diet and infected or not with *A. hydrophila*. Data are presented as mean ± SEM (*n* = 8). Two-way ANOVA and Tukey’s test, *p* < 0.05. Different lowercase letters indicate significant differences between healthy groups. Different capital letters indicate a significant difference between groups infected with *A. hydrophila*. The number sign (#) indicates a significant difference between the groups infected with *A. hydrophila* in relation to the healthy groups. The asterisk (*) indicates a significant difference between the groups infected with *A. hydrophila* in relation to the respective healthy control group.

**Figure 5 animals-13-03307-f005:**
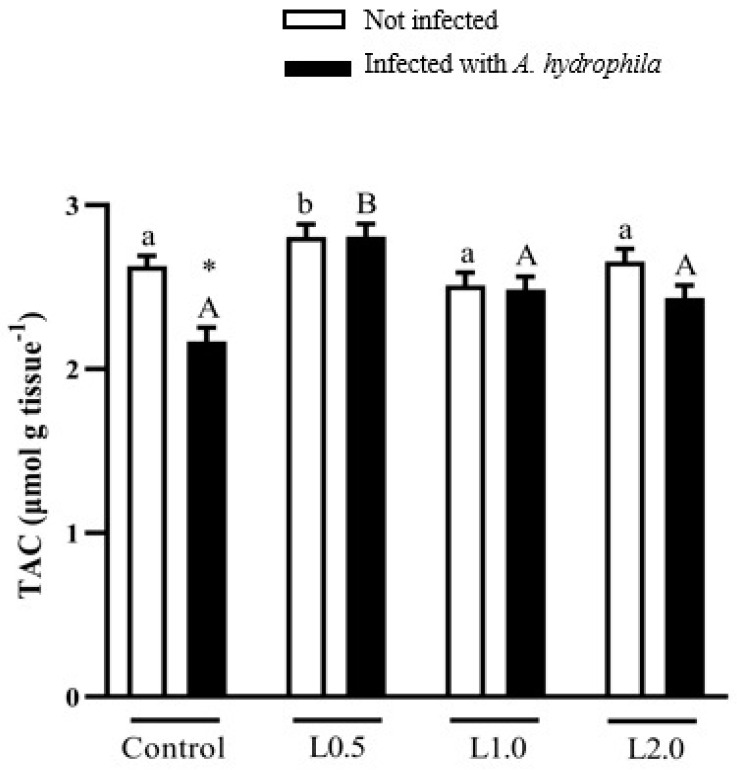
Total antioxidant capacity (TAC) in the liver of *Rhamdia quelen* fed with different doses of R-(+)-limonene in the diet and infected or not with *A. hydrophila*. Data are presented as mean ± SEM (*n* = 8). Two-way ANOVA and Tukey’s test, *p* < 0.05. Different lowercase letters indicate significant differences between healthy groups. Different capital letters indicate a significant difference between groups infected with *A. hydrophila*. The asterisk (*) indicates a significant difference between the groups infected with *A. hydrophila* in relation to the respective healthy control group.

**Figure 6 animals-13-03307-f006:**
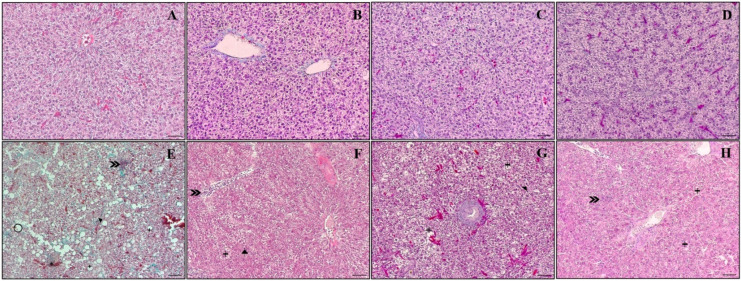
Effects of R-(+)-limonene supplemented in the diet on histopathological findings observed in the liver of *Rhamdia quelen* infected or not with *A. hydrophila*. Representative images of *Masson-Goldner* trichrome histological staining in the liver tissue of silver catfish exposed to R-(+)-limomene. Control (**A**), L0.5 (**B**), L1.0 (**C**), L2.0 (**D**), without *A. hydrophila*; control (**E**), L0.5 (**F**), L1.0 (**G**), L2.0 (**H**), with *A. hydrophila.* (*n* = 8). 200× (bar = 20 µm). + (vacuolation of hepatocytes), 
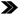
 (infiltration of erythrocytes into the blood sinusoids), 
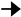
 (nuclear pyknosis), 
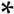
 (hemorrhage), 
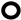
 (fibrosis), 
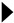
 (binucleated hepatocytes).

**Figure 7 animals-13-03307-f007:**
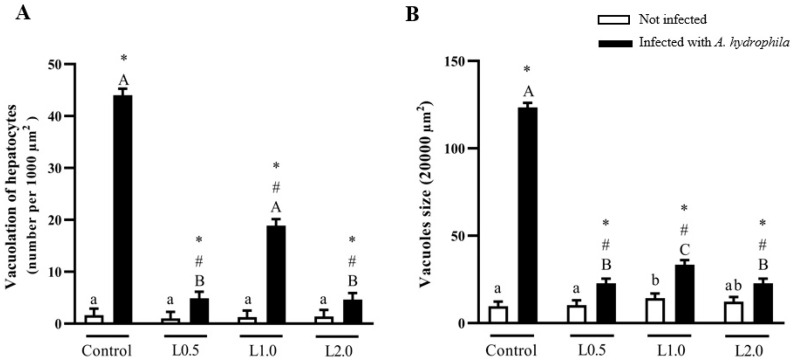
Quantitative (**A**) and morphometric (**B**) analysis of vacuoles in hepatocytes in the liver of *Rhamdia quelen* fed different doses of R-(+)-limonene in the diet and infected or not with *A. hydrophila*. Data are presented as mean ± SEM (*n* = 8). Two-way ANOVA and Tukey’s test, *p* < 0.05. Different lowercase letters indicate significant differences between healthy groups. Different capital letters indicate significant differences between groups infected with *A. hydrophila*. The number sign (#) indicates a significant difference between the groups infected with *A. hydrophila* in relation to the healthy groups. The asterisk (*) indicates a significant difference between the groups infected with *A. hydrophila* in relation to the respective healthy control group.

**Figure 8 animals-13-03307-f008:**
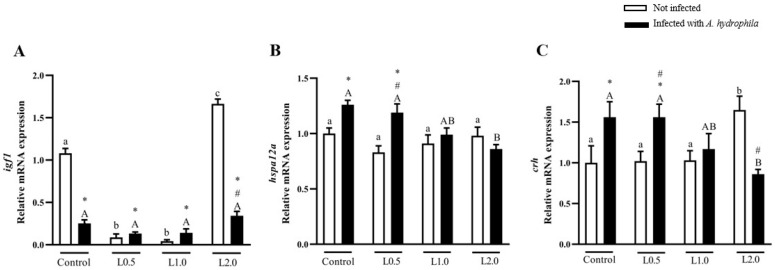
Gene expression of *igf1* in the liver (**A**), and *hsp12a* (**B**) and *crh* (**C**) in the brain of *Rhamdia quelen* fed different doses of R-(+)-limonene in the diet and infected or not with *A. hydrophila*. Data are presented as mean ± SEM (*n* = 8). Two-way ANOVA and Tukey’s test, *p* < 0.05. Different lowercase letters indicate significant differences between healthy groups. Different capital letters indicate a significant difference between groups infected with *A. hydrophila*. The number sign (#) indicates a significant difference between the groups infected with *A. hydrophila* in relation to the healthy groups. The asterisk (*) indicates a significant difference between the groups infected with *A. hydrophila* in relation to the respective healthy control group.

**Table 1 animals-13-03307-t001:** Formulation (%) of the experimental standard diet.

Ingredients	(g/kg)
Meat meal	350
Soybean meal	300
Corn	150
Rice bran	120
Canola oil	30
Salt	10
Vitamin and mineral (pré-mix) *	30
Proximate composition	
Dry matter	939.9
Mineral matter	252.3
Crude protein	303.2
Ether extract	74.9
Neutral detergent fiber	251.2
Acid detergent fiber	7.8

* Vitamin and mineral mixture (security levels per kilogram of product)-folic acid: 250 mg; pantothenic acid: 5000 mg; antioxidant: 0.60 g; biotin: 125 mg; cobalt: 25 mg; copper: 2000 mg; iron: 820 mg; iodide: 100 mg; manganese: 3750 mg; niacin: 5000 mg; selenium: 75 mg; vitamin A: 1,000,000 UI; vitamin B1: 1250 mg; vitamin B12: 3750 mcg; vitamin B2: 2500 mg; vitamin B6: 2485 mg; vitamin C: 28,000 mg; vitamin D3: 500,000 UI; vitamin E: 20,000 UI; vitamin K: 500 mg; and zinc: 17,500 mg.

**Table 2 animals-13-03307-t002:** Growth performance of silver catfish (*Rhamdia quelen*) fed diets containing different doses of the R-(+)-limonene isoform.

Growth Performance	R-(+)-Limonene Concentration (mL/kg Diet)			
	0 (control)	L (0.5)	L (1.0)	L (2.0)
Initial				
W	20.21 ± 0.07	19.21 ± 0.04	19.81 ± 0.02	20.07 ± 0.04
SL	11.39 ± 0.02	10.55 ± 0.05	11.07 ± 0.04	11.12 ± 0.02
30 days				
W	31.91 ± 1.46 ^a^	41.78 ± 1.35 ^b^	32.50 ± 1.46 ^ab^	44.03 ± 1.40 ^b^
SL	14.28 ± 0.40 ^a^	14.86 ± 0.40 ^a^	14.70 ± 0.40 ^a^	15.07 ± 0.40 ^a^
SGR	1.78 ± 0.21 ^a^	3.58 ± 0.17 ^b^	3.14 ± 0.18 ^a^	4.29 ± 0.18 ^b^
60 days				
W	62.24 ± 1.06 ^a^	88.44 ± 1.50 ^b^	73.33 ± 1.11 ^a^	91.33 ± 1.73 ^b^
SL	17.10 ± 0.41 ^a^	18.17 ± 0.41 ^a^	17.68 ± 0.40 ^a^	19.44 ± 0.40 ^a^
FI	66.43 ± 1.11 ^a^	75.77 ± 1.30 ^a^	68.03 ± 1.40 ^a^	68.84 ± 1.10 ^a^
FCR	1.79 ± 0.02 ^a^	1.33 ± 0.02 ^bc^	1.83 ± 0.04 ^a^	1.11 ± 0.05 ^c^
SGR	3.45 ± 0.07 ^a^	3.99 ± 0.05 ^c^	3.67 ± 0.07 ^b^	3.98 ± 0.07 ^c^
HSI	1.29 ± 0.10 ^a^	1.21 ± 0.10 ^a^	1.27 ± 0.11 ^a^	1.18 ± 0.11 ^a^
Survival	100	100	100	100

Data are presented as mean ± SEM (*n* = 9). Different letters in the rows indicate a significant difference between treatments (one-way ANOVA and Tukey Test, *p* < 0.05). Abbreviations: weight (W) (g), standard length (cm) (SL), feed intake (%) (FI), feed conversion rate (FCR), specific growth rate (%) (SGR), hepatosomatic index (%) (HSI), and survival (%).

**Table 3 animals-13-03307-t003:** Metabolic parameters of silver catfish *Rhamdia quelen* fed with diets containing different doses of R-(+)-limonene.

R-(+)-Limonene Dose (mL/kg/Diet)				
Non-Infected Groups				
**Metabolites**	**0 (Control)**	**L (0.5)**	**L (1.0)**	**L (2.0)**
Plasma				
Glucose	72.32 ± 5.89 ^a^	79.80 ± 5.51 ^a^	75.37 ± 5.00 ^a^	79.55 ± 5.00 ^a^
Lactate	35.35 ± 2.38 ^a^	26.08 ± 2.38 ^a^	27.29 ± 2.55 ^a^	27.67 ± 2.38 ^a^
Total Protein	5.08 ± 0.34 ^a^	5.30 ± 0.32 ^a^	4.55 ± 0.32 ^a^	4.53 ± 0.32 ^a^
LDH	154.45 ± 10.82 ^a^	135.99 ± 10.82 ^a^	138.10 ± 10.82 ^a^	119.77 ± 10.82 ^b^
Liver				
Glucose	56.44 ± 2.92 ^a^	56.44 ± 2.92 ^a^	64.69 ± 3.46 ^ab^	72.50 ± 3.43 ^b^
Lactate	21.89 ± 3.12 ^a^	21.89 ± 3.12 ^a^	20.03 ± 3.12 ^a^	24.09 ± 3.12 ^a^
Glycogen	41.69 ± 2.5 ^a^	35.08 ± 2.50 ^a^	38.83 ± 2.50 ^a^	37.05 ± 2.50 ^a^
Total Protein	126.51 ± 3.64 ^a^	148.34 ± 3.76 ^a^	131.11 ± 3.89 ^a^	137.77 ± 3.48 ^a^
Groups Infected with *A. hydrophila*				
Plasma				
Glucose	123.88 ± 6.36 ^A^*	67.70 ± 6.36 ^B^	57.51 ± 5.00 ^B^	60.10 ± 6.00 ^B^
Lactate	68.20 ± 2.55 ^A^*	53.46 ± 2.38 ^B#^*	36.70 ± 2.38 ^C^	39.78 ± 2.38 ^C#^
Total Protein	8.10 ± 0.34 ^A^*	4.72 ± 0.32 ^B^	3.93 ± 0.32 ^B^	4.67 ± 0.32 ^B^
LDH	230.08 ± 9.14 ^A^*	131.20 ± 10.82 ^B^	94.36 ± 10.82 ^B#^*	137.03 ± 10.82 ^B^
Liver				
Glucose	112.73 ± 2.90 ^A^*	65.02 ± 3.43 ^B^	66.84 ± 3.13 ^B^	67.47 ± 3.13 ^B^
Lactate	68.95 ± 3.12 ^A^*	30.99 ± 3.12 ^B#^	27.25 ± 3.12 ^C^	29.16 ± 3.12 ^BC^
Glycogen	134.22 ± 2.50 ^A^*	42.91 ± 2.50 ^B#^	55.28 ± 2.50 ^C#^*	60.24 ± 2.50 ^C#^*
Total Protein	109.69 ± 4.12 ^A^	152.21 ± 3.26 ^B^*	151.16 ± 3.48 ^B^	148.70 ± 3.48 ^B^

Data are presented as mean ± SEM (*n* = 9). Two-way ANOVA and Tukey’s test, *p* < 0.05. Plasma: glucose and lactate (mg/dL), protein (g/dL), LDH (U/L). Liver: glucose, lactate, glycogen, and protein levels were expressed as μmol/g of tissue. Different lowercase letters indicate significant differences between healthy groups. Different capital letters indicate a significant difference between groups infected with *A. hydrophila*. The asterisk (*) indicates a significant difference from the respective healthy group. The number sign (^#^) indicates a significant difference between groups with *A. hydrophila.*

**Table 4 animals-13-03307-t004:** Lipid profile of silver catfish *Rhamdia quelen* fed with diets containing different doses of R-(+)-limonene.

R-(+)-Limonene Dose (mL/kg/Diet)				
Non-Infected Groups				
**Lipid Profile**	**0 (Control)**	**L (0.5)**	**L (1.0)**	**L (2.0)**
Plasma				
Triglycerides	445.04 ± 28.06 ^a^	396.45 ± 28.06 ^a^	268.30 ± 30.31 ^b^	294.66 ± 30.31 ^b^
Cholesterol	111.36 ± 4.49 ^a^	113.52 ± 5.31 ^a^	106.09 ± 5.94 ^a^	89.75 ± 5.31 ^a^
Liver				
Total lipid	786 ± 25.34 ^a^	618.67 ± 22.66 ^b^	703.10 ± 22.66 ^a^	744.32 ± 25.34 ^a^
Groups Infected with *A. hydrophila*				
Plasma				
Triglycerides	918.64 ± 33.20 ^A^*	256.69 ± 30.31 ^B#^*	228.75 ± 30.31 ^B^*	301.37 ± 28.06 ^B^*
Cholesterol	219.85 ± 5.31 ^A^*	105.84 ± 4.85 ^B^	111.20 ± 5.31 ^B^	95.69 ± 4.85 ^B^
Liver				
Total lipid	909.48 ± 22.66 ^A^*	925.76 ± 25.34 ^A#^*	719.33 ± 22.66 ^B^	712.04 ± 25.34 ^B^

Data are presented as mean ± SEM (*n* = 9). Two-way ANOVA and Tukey’s test, *p* < 0.05. Plasma: triglycerides (mg/dL), cholesterol (mg/dL). Liver: total lipids (mg/g tissue). Different lowercase letters indicate significant differences between healthy groups. Different capital letters indicate a significant difference between groups infected with *A. hydrophila*. The asterisk (*) indicates a significant difference from the respective healthy group. The number sign (^#^) indicates a significant difference between groups with *A. hydrophila.*

## Data Availability

Data are available upon request to the authors.

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
