# Peer review of "Dietary Supplementation with R-(+)-Limonene Improves Growth, Metabolism, Stress, and Antioxidant Responses of Silver Catfish Uninfected and Infected with Aeromonas hydrophila"

_animals, 2023, doi:10.3390/ani13213307_

Round 1

Reviewer 1 Report (Previous Reviewer 1)

I have no further comments. The manuscript was very carefully revised.

Reviewer 2 Report (Previous Reviewer 3)

This manuscript has been corrected by the Authors in a very good way and can be published in its present form in my opinion. Well done!

Reviewer 3 Report (Previous Reviewer 4)

The authors edited their manuscript paragraph by paragraph. They answered all reviewers' questions and comments. In this form the manuscript significantly improved and can now be accepted for publication in the journal Animals.

This manuscript is a resubmission of an earlier submission. The following is a list of the peer review reports and author responses from that submission.

Round 1

Reviewer 1 Report

 Animal Welfare in cultured aquatic organisms is a very relevant topic and also the public is aware about. Studies which contribute in this field are of special interest. Essential oils are known to positively affect fish welfare and performance.

In the present study the influence of limonene on performance of silver catfish and its resistance against Aeromonas infection is investigated.

The data extend the existing knowledge, have an actuality and are of importance for the field of aquaculture.

Before publication of the manuscript, I recommend a major revision. The authors are requested to consider the following points:

The study contains novel parts which concern the effect of limonene on resistance in silver catfish. However, the study contains also parts, which are very similar to a previous study (i.e. reference 16). The effects of limonene concentrations on body indices, liver energy parameters and liver antioxidant system seem a reinvestigation of the previous study. Also the experimental design is similar in the two study. I recommend the authors to delete or shorten these data (with exception of those needed as controls for the infection experiments). A clear demarcation from the previous study is necessary.

  Histology: The evaluated histological parameters might need morphometric analysis. It is unclear if the evaluated characteristics (vacuolation of hepatocytes/lipid deposition in hepatocytes, etc.) are present in all infected fish. Also the frequency of these characteristics cannot be evaluated from the micrographs. The meaning of mean ± SEM (n = 8) for histological images is unclear. Conclusion in lines 606/607 of discussion can only be drawn when histological features are investigated quantitatively

Authors are requested to indicate effectiveness of infection (infection rate). Could the authors observe any external symptoms or is infection only manifested in liver histology? In this case liver of each fish should be investigated in the end of the experiment for pathological structures.

 In material and methods the description of sample processing is necessary, i.e. the stabilization of tissue in buffers and or storage and the extraction of the specific compounds.

 For several analytes it is not clearly stated where they were measured: This concerns compounds listed in chapters  2.6.6. – 2.6.10. Also for transaminases it unclear if they have been measured in serum (as diagnostic markers for tissue damage and leakage out of specific organs) or as activity in the liver.

 In all Figures the legend “with Aeromonas” and “without Aeromonas” seems confused.

Specific and minor comments:

 Title: please improve: the sense is unclear and does not clearly reflect the content of the paper, metabolic is adverb and should be replaced by metabolism

 Line 17: the term metabolism should be defined more precisely; a big part of the study is related to energy metabolism and to antioxidant systems

 Line 39 – “some” better to say harmful or suboptimal conditions

 Line 40/41 the sentence is unclear

 line 95 Please indicate how Aeromonas was cultured or isolated. Pure strains seem essential for infection

 Results: I think growth performance data after 30 days are not necessary. The main intention of the study is the comparison of parameters in uninfected fish versus Aeromonas infected fish at different limonene concentrations

 In table 3 liver glucose, lactate, protein, glycogen is a reinvestigation of a previous study. Liver total protein value (41.69±2.5) seems wrong, a copy paste failure as it is identical with glycogen

Tables 3, 4: Concerning the organization, the tables are difficult to read. Better organization should be preferred

 In Fig 8A IgfI is downregulated in limonene (0.5., 1.0) fed fish, however growth is similar or enhanced in comparison to control. Can this finding be discussed in more details?  

 Line 439/443: I am not sure that plasma protein is a sensitive parameter to characterize immunity of fish (please refer to literature on cellular and molecular immune parameters in fish). Moreover, the conclusion is unclear: Do you assume that immunity is boosted or inhibited by high plasma protein levels? And: plasma protein is no energy resource in fish

 Line 498/499: Generally, glucose is liberated from glycogen depots in the liver under stress and not reverse

 Iine 50: do you have indications for protein catabolism in liver during stress - or references? this seems speculative

 Discussion in general: it is often unclear how authors evaluate the increase or decrease of a specific parameter. They do not say if it is a positive or negative indication for fish metabolism, performance, or health. The discussion could benefit from clear statements.

 A general question about the use of limonene: Can this essential change the taste of fish and lead to problems in product quality?

Reviewer 2 Report

This manuscript entitled “ Dietary Supplementation With R-(+)-limonene Improves Growth and Metabolic, Stress, and Antioxidant Responses of Silver Catfish Infected With Aeromonas hydrophila ” mainly explored the effect of dietary R-(+)-limonene supplementation on growth, metabolic response, antioxidant property, and expression of related gene in silver catfish (Rhamdia quelen) with Aeromonas hydrophila infection. Findings from this study could highlight the importance of citrus plants-derived essential oil compounds to improve growth performance and health of silver catfish and other fish species, particularly in the case of bacterial infections and provide the technical support for developing and optimizing safe, environment-friendly feed additives in aquaculture.

Findings in this manuscript are meaningful. In this case, the paper should be carefully proofread throughout for language to eliminate several typos and grammatical errors.

Major comments:

1. In Line 49, the phrase "oxidative stress indicators" is very confusing. For some oxidative indicators, such as MDA and SOD, the higher concentration of MDA is related to high levels of oxidative stress response, while higher SOD activity is helpful to fight against oxidative stress. So what does the statement “oxidative stress indicators” exactly mean? Does it mean "improve or enhance the antioxidant capability" or "reduce the antioxidant capability" ?

2. In "2.6.12. Gene expression" section, the description "n=8 per group for all the expressed genes" (Line 164-165) indicates three target genes or only igf1 gene? Please rephrase it for a clearer presentation.

3. The results contents in this manuscript ("4. Results" section, Page 5-13) should be simplified and described with emphasis.

4. Regarding the figure, we propose to integrate Figure 3 and Figure 4 into a single figure for a more clear representation of glutathione-associated result in this paper.

5. Generally, carbohydrates, lipids, and proteins are the three major energy sources mobilized by animals, including fish. But in Line 442, the statement "Glucose is another source of energy for fish, in addition to protein" seemed to indicate that there are only two energy sources for fish. The authors should explain this statement more clearly or rephrase it more precisely.

Minor comments:

1. There are too many keywords in this manuscript (Line 29-30). Please compress into 4-6 keywords.

2. Check the symbols or codes for volume unit in "2.3 Diets" section according to the information to related guides. In Line 82, please replace "ml" with "mL".

3. Both "analyse" (Table 1, Page 3) and "analyze" (Line 135) are in use. Which one is correct in this manuscript? Please revise the same errors in this paper.

Other errors were presented in the PDF file.

Therefore, this manuscript will be reconsidered after major revision.

This manuscript entitled “ Dietary Supplementation With R-(+)-limonene Improves Growth and Metabolic, Stress, and Antioxidant Responses of Silver Catfish Infected With Aeromonas hydrophila ” mainly explored the effect of dietary R-(+)-limonene supplementation on growth, metabolic response, antioxidant property, and expression of related gene in silver catfish (Rhamdia quelen) with Aeromonas hydrophila infection. Findings from this study could highlight the importance of citrus plants-derived essential oil compounds to improve growth performance and health of silver catfish and other fish species, particularly in the case of bacterial infections and provide the technical support for developing and optimizing safe, environment-friendly feed additives in aquaculture.

Findings in this manuscript are meaningful. In this case, the paper should be carefully proofread throughout for language to eliminate several typos and grammatical errors.

Major comments:

1. In Line 49, the phrase "oxidative stress indicators" is very confusing. For some oxidative indicators, such as MDA and SOD, the higher concentration of MDA is related to high levels of oxidative stress response, while higher SOD activity is helpful to fight against oxidative stress. So what does the statement “oxidative stress indicators” exactly mean? Does it mean "improve or enhance the antioxidant capability" or "reduce the antioxidant capability" ?

2. In "2.6.12. Gene expression" section, the description "n=8 per group for all the expressed genes" (Line 164-165) indicates three target genes or only igf1 gene? Please rephrase it for a clearer presentation.

3. The results contents in this manuscript ("4. Results" section, Page 5-13) should be simplified and described with emphasis.

4. Regarding the figure, we propose to integrate Figure 3 and Figure 4 into a single figure for a more clear representation of glutathione-associated result in this paper.

5. Generally, carbohydrates, lipids, and proteins are the three major energy sources mobilized by animals, including fish. But in Line 442, the statement "Glucose is another source of energy for fish, in addition to protein" seemed to indicate that there are only two energy sources for fish. The authors should explain this statement more clearly or rephrase it more precisely.

Minor comments:

1. There are too many keywords in this manuscript (Line 29-30). Please compress into 4-6 keywords.

2. Check the symbols or codes for volume unit in "2.3 Diets" section according to the information to related guides. In Line 82, please replace "ml" with "mL".

3. Both "analyse" (Table 1, Page 3) and "analyze" (Line 135) are in use. Which one is correct in this manuscript? Please revise the same errors in this paper.

Other errors were presented in the PDF file.

Therefore, this manuscript will be reconsidered after major revision.

Reviewer 3 Report

Major comments

In my opinion, the article is interesting but the experiment was conducted with methodological errors (please see 1). Moreover, the manuscript contains numerous inaccuracies:

1) Biochemistry. The blood centrifugation parameters used (only 1000 g) are not suitable for obtaining platelet-free or platelet-poor plasma. The performed biochemical determinations are therefore burdened with a large error (large uncertainty of the results) and it is the authors' duty to write about it directly in the Discussion.

2) Water chemistry. Nitrate concentrations are not tested, fortunately the nitrite level has been determined and is very low. This does not change the fact that with a relatively small water change (only 20% per day) ammonia, nitrites and nitrates should be controlled. The authors should therefore briefly justify (explain) in the Material and Methods why they did not test the concentration of nitrates.

3) The authors stated: “In plasma, alanine aminotransferase (ALT) and aspartate aminotransferase (AST) were measured using kits (Gold Analisa, Belo Horizonte, Brazil) and lactate (BioTecnica).” Was ALT / AST concentration or activity measured? It should be clarified in the manuscript.

4) The authors did not specify which two factors were tested using two-way ANOVA and what are the results of this analysis. They only provided the results of the post hoc analysis. Therefore, I am not sure whether post hoc could be carried out at all (of course, post hoc in the case of two-way ANOVA can only be done when the significance of the interaction of both factors has been demonstrated). Please, add the results of one-way and two-way ANOVA (p-values), not only results of post hoc analyses. It is especially important in the case of two-way model and eventually can be omitted in one-way model.

5) The performance of the ANOVA test requires the fulfilment of two assumptions: compliance of the obtained results with the normal distribution and homogeneity of variance. The authors did not indicate whether the compliance of their data with the normal distribution was examined. Why?

6) The authors mention that they performed Levene's test but did not report what the results of this testing were. It does not make sense. You might as well write that Tukey's test was performed, but without providing its result in the Results content and in the tables.

What was the test result? Was the p-value in the Levene’s test higher than the significance level (0.05) in most or in all of the analysed cases? I am asking the authors to complete this information.

Minor comments

The following issues also should be clarified:

1) The authors use the word “concentration” (of limonene) for the applied substance but the word “concentration” refers, by its definition, to aqueous solutions. Wouldn't it be better to use a different term? Maybe “content” would be good? Please think about this issue.

2) If the tests were performed in triplicate (as stated in line 101), how were the results analyzed? Were they averaged (into 4 groups) or were 12 groups analyzed? Please provide a detailed description in the “Statistical analysis” section.

3) In the “Growth performance” section, the Authors provide many formulas. What is their literary source? Please add if possible.

4) "p<0.05" is repeated many times in the Results section. When the p-value obtained during the statistical analysis was less than 0.01 or less than 0.001, wouldn't it be worth to inform the reader about such cases? Weren't there any such cases? Or maybe the Authors did not record such cases? Of course, this could have happened, but the reader does not know about it.

5) Line 595–598: “Mustahal et al. [87] observed that untreated catfish (Clarias sp.) infected with A. hydrophila exhibited hepatic congestion and lipid accumulation in the liver by formation of vacuoles, which according to Kalaiyarasi et al. [88], compromises the lipid metabolism of fish”

Untreated with what? I don’t understand. With chemicals?

6) Line 611–612: “HSP are molecules that can be activated in situations of stress and in the immune response in bacterial infections, acting to protect proteins against denaturation caused by stressors [92]”. Please explain the abbreviation “HSP” if possible.

7) Line 619: HPI or HPA? Or maybe both HPI and HPA? I am not sure, please think about this issue. Of course, it is only a suggestion, this may not be the correct hint.

8) The Authors took beautiful histological pictures, but they are so small that you can't see much on them. I suggest considering larger photos of histopathological slides.

9) Contributions of all the Authors should be added.

10) The introduction is a bit too short. The authors should better discuss the role of various dietary supplements and immunostimulants used in fish farming. Please refer to papers of this type, for example, the following ones can be used:

Morshedi, Vahid, et al. "Effects of dietary bovine lactoferrin on growth performance and immuno-physiological responses of Asian sea bass (Lates calcarifer) fingerlings." Probiotics and Antimicrobial Proteins 13.6 (2021): 1790-1797.

Tangestani, N., Nafisi, M., Morshedi, V., Bagheri, D., Sotoudeh, E., Ghasemi, A., & Bojarski, B. (2023). Effects of Dietary Macroalgae Gracilaria pygmaea on Asian Sea Bass (Lates calcarifer) Juveniles. Journal of Agricultural Science and Technology, 647-660.

Abdelnour, S. A., Ghazanfar, S., Abdel-Hamid, M., Abdel-Latif, H. M., Zhang, Z., & Naiel, M. A. (2023). Therapeutic uses and applications of bovine lactoferrin in aquatic animal medicine: an overview. Veterinary Research Communications, 1-15.

11) Line 51 (Introduction) – Please add:

“Blood biochemical parameters are markers commonly used in aquaculture and scientific research to evaluate the health and well-being of fish (Seibel et al. 2021; Witeska et al. 2022). However, for a comprehensive assessment of the effects of various environmental or dietary factors, physiological analyses should be supplemented by histological ones (e.g. Figueiredo‐Silva et al. 2005; Rašković et al. 2011; Lozano et al. 2017; Bojarski et al. 2022).

Lozano, A. R., Borges, P., Robaina, L., Betancor, M., Hernandez-Cruz, C. M., García, J. R., ... & Izquierdo, M. (2017). Effect of different dietary vitamin E levels on growth, fish composition, fillet quality and liver histology of meagre (Argyrosomus regius). Aquaculture468, 175-183.

Witeska, M., Kondera, E., Ługowska, K., & Bojarski, B. (2022). Hematological methods in fish–Not only for beginners. Aquaculture547, 737498.

Seibel, H., Baßmann, B., & Rebl, A. (2021). Blood will tell: what hematological analyses can reveal about fish welfare. Frontiers in Veterinary Science8, 616955.

Bojarski, B., Osikowski, A., Hofman, S., Szała, L., Szczygieł, J., & Rombel-Bryzek, A. (2022). Effects of exposure to a glyphosate-based herbicide on haematological parameters, plasma biochemical indices and the microstructure of selected organs of the common carp (Cyprinus carpio Linnaeus, 1758). Folia Biologica (Kraków)70(4), 213-229.

Rašković, B., Stanković, M., Marković, Z., & Poleksić, V. (2011). Histological methods in the assessment of different feed effects on liver and intestine of fish. Journal of Agricultural Sciences (Belgrade)56(1), 87-100.

FigueiredoSilva, A., Rocha, E., Dias, J., Silva, P., Rema, P., Gomes, E., & Valente, L. M. P. (2005). Partial replacement of fish oil by soybean oil on lipid distribution and liver histology in European sea bass (Dicentrarchus labrax) and rainbow trout (Oncorhynchus mykiss) juveniles. Aquaculture Nutrition11(2), 147-155.

Reviewer 4 Report

The paper “Dietary Supplementation With R-(+)-limonene Improves Growth and Metabolic, Stress, and Antioxidant Responses of Silver Catfish Infected With Aeromonas hydrophila” describes the results of a comprehensive experimental study important for fish farming practice.

The experimental design is set up correctly. Fish in groups are standardized in size and weight. Fish groups for each individual test are sufficient in number for statistical analysis. Before the experiment, the fish were acclimated for seven days.  Fish density in tanks were acceptale. Conditions were with continuous aeration. Dissolved oxygen, pH level, temperature, ammonia and nitrite levels were controlled. The food of fish was standardized. Survival in all fish groups by the end of the experiment were 100%, which indicates good conditions of keeping the fish.

Ethical standards for the use of animals are respected.

The description of the experiment is complete and detailed. Diet components are described in details. A standardized procedure of infecting fish with A. hydrophila is clearly described.

A control and 3 concentrations of limonene in food were used. Growth patameters were checked on 30th and 60th days. On day 60, the fish were infected with A. hydrophila for one week. After this period, the fish were anesthetized for blood sampling and then euthanized to separate the liver and brain tissues.

Thus, it can be concluded that the obtained results are correct. There are no questions about their objectivity.

However, there are a few suggestions that, in my opinion, can improve the understanding and better structure the results presented.

1). The introduction is incomplete. The aim of the work (lines 58-60) should be supplemented by the objectives of the study.

Something like this:

The objectives of the study were a comprehensive analysis of fish growth parameters, as well as a number of metabolic parameters of blood plasma and liver, including indicators of lipid metabolism and the antioxidant system. The focus was on liver histology and function, a key organ in metabolic processes. Brain tissue was selected to study the expression of genes related to infection stress responses.

2). The part “Results” describes a lot of parameters from 4.1 to 4.12.

However, there are two cross-cutting themes in all of them: 1) the effects of food additives on the growth and metabolism of healthy (uninfected) catfish and 2) the effects of food additives on the resistance of fish to infection stressor of A. hydrophila. They are equally important, and it seems that each can have its own group of interested readers. Therefore, each of two should be summarized briefly at the end of the chapter Results:

1) Positive effects of dietary supplementation on healthy catfish (one paragraph)

2). Positive effects of dietary supplementation on fish resistance to infection stressors (one paragraph)

And each of two should be emphasized throughout the other parts of the text  (for example, see comments to the Title - in the file attached).

The MS can be publish with minor corrections
